# The Paradox of Sexual Dysfunction Observed during Pregnancy

**DOI:** 10.3390/healthcare11131914

**Published:** 2023-07-03

**Authors:** Ana-Maria Cristina Daescu, Dan-Bogdan Navolan, Liana Dehelean, Mirela Frandes, Alexandru-Ioan Gaitoane, Andrei Daescu, Razvan-Ionut Daniluc, Dana Stoian

**Affiliations:** 1Doctoral School Department, Victor Babes University of Medicine and Pharmacy, 300041 Timisoara, Romania; 2Department of Internal Medicine II, Victor Babes University of Medicine and Pharmacy, 300041 Timisoara, Romania; 3Neurosciences Department, Victor Babes University of Medicine and Pharmacy, 300041 Timisoara, Romania; 4Department of Obstetrics and Gynecology, Victor Babes University of Medicine and Pharmacy, 300041 Timisoara, Romania; 5Functional Sciences Department, Victor Babes University of Medicine and Pharmacy, 300041 Timisoara, Romania; 6Department of Mathematics, Politehnica University of Timisoara, 300006 Timisoara, Romania; alexandru.gaitoane@student.upt.ro (A.-I.G.); andrei.daescu2@student.upt.ro (A.D.)

**Keywords:** pregnancy, female sexual dysfunction, FSFI, BESAQ, BDI-II

## Abstract

The aim of this study is to analyze the evolution of sexual function throughout pregnancy and highlight the predicting factors of sexual dysfunction in pregnant women. Our study included 144 participants, aged 16 to 45. Patients were evaluated three times during pregnancy by filling out the Female Sexual Function Index (FSFI) and the Body Exposure in Sexual Activities Questionnaire (BESAQ). At the time of the last evaluation, we asked patients to also fill out Beck’s Depression Inventory (BDI-II) and a questionnaire regarding their psychological status and relationship satisfaction. We observed that the FSFI lubrication, satisfaction, and pain domains and the FSFI total score significantly decreased from the 1st to the 3rd evaluation. We observed that an increase in BDI score and the presence of abortion in the patient’s history increase the risk of developing female sexual dysfunction (FSD). Higher BMIs were found to be a protective factor against FSD, as was being unmarried. The relationship satisfaction score was found to be an independent predictor of FSD. These findings support previous studies that indicate that pregnancy and postpartum sexuality are multifaceted phenomena and that psycho-social factors have a greater impact on sexuality than biological factors.

## 1. Introduction

Sexual function is a fundamental component of life and is considered part of a woman’s health [1]. According to the World Health Organization (WHO), sexual health includes physical, mental, emotional, and social well-being in all sexual behaviors and beliefs [2]. Female sexual function contributes to women’s quality of life, with studies in the field showing that healthy sexual activities and sexual satisfaction are associated with better psychological and physical function, with beneficial effects on metabolism, analgesia, cardiovascular health, immune function, and psychological well-being [3].

Female sexual dysfunction (FSD) is a highly prevalent condition that affects up to 40–50% of women globally [4]. FSD is described as a woman’s inability to participate in a sexual activity as she would wish and experimenting with a persistent or recurring decrease in sexual desire or arousal, the difficulty of achieving an orgasm, and the feeling of pain during sexual intercourse [5,6]. The Diagnostic and Statistical Manual of Mental Disorders, 5th edition (DSM-5) criteria for female sexual dysfunction require that the sexual symptoms be present for a minimum duration of approximately 6 months, causing clinically significant distress in the individual [7].

Even though it is essential, there is not enough dialogue between women and healthcare professionals on female sexual health [8]. Women are not discussing their concerns with their doctors, and doctors are not routinely including sexual health screenings in their examinations [9]. Women’s health care providers lack confidence in addressing sexual health issues and often underestimate the prevalence of female sexual dysfunction. This may result in negative physical, psychological, and emotional consequences [10].

Pregnancy is an especially delicate time to address sexual health issues [11]. The WHO has suggested that research on sexual health needs to be performed and has consistently highlighted the need to provide prenatal and postpartum care, information, and counseling for women [2].

Female sexual function and sexual health during pregnancy are key areas of concern for women’s health care providers. Women go through physical, hormonal, psychological, emotional, and social changes that influence sexual behavior during pregnancy [12,13]. Changes that occur at each stage of pregnancy may have a substantial influence on couples’ sexual lives. Generally, sexual function declines during pregnancy and remains low during the postpartum period [14]. In the literature, it has been shown that there is an association between pregnancy and sexual dysfunction [15]. In the first trimester of pregnancy, sexual desire frequently diminishes because of exhaustion, nausea, emotional lability, and increased anxiety or fears about miscarriage [16]. In the second trimester, there is an increase in libido as physical symptoms decrease, vaginal lubrication improves, and earlier anxiety diminishes as women psychologically adapt to pregnancy [16]. In the third trimester, physical difficulties women experience usually make sexual activities more uncomfortable and less frequent [16]. Studies on female sexual function during pregnancy showed that throughout the first trimester, 96% of pregnant women participated in vaginal intercourse, but by the third trimester, just 67% did [17]. Prior to becoming pregnant, 84.7% of women had one to four times per week of sexual intercourse; this dropped to 70% in the first trimester, 61.3% in the second, and 32% in the third trimester [18]. During the third trimester, 52% to 73% of women meet clinical criteria for sexual dysfunction [19]. In the first, second, and third trimesters of pregnancy, respectively, 66.3%, 50.7%, and 69.2% of women experienced sexual dysfunction, with sexual desire disorder being the most often reported sexual dysfunction in each trimester [20]. Studies found that during pregnancy, just 11.2% of pregnant women exhibited favorable attitudes towards sexuality [21]. Also, 68% of women who were surveyed admitted they had never brought up sexual issues with their doctor while they were pregnant [22].

Evidence shows that women with sexual dysfunction often experience problems of low self-esteem, emotional distress, anxiety, and symptoms of depression [23,24]. Some stressors that cause anxiety and depression have been investigated as variables influencing sexual health during pregnancy [24,25]. Other stressors affecting female sexual function were found to be the fear of harming the baby through sexual activities, stress concerning labor and delivery, a lack of support from the partner, increased responsibility associated with having a baby, and not having as much independence [26].

One of the most important variables in marital happiness and the overall quality of life of couples is sexual satisfaction. Issues impacting sexual activity can lead to marital conflicts and decreased relationship satisfaction. Also, marital stress and conflict may lead to different pregnancy complications and have long-term consequences for newborns’ and couples’ lives [27]. As part of prenatal care, pregnant women’s sexual health and sexual issues should be examined [28]. Examining sexual function in pregnant women might help to improve the sexual health of couples and, implicitly, the quality of their relationships and lives. The aim of this study is to analyze the evolution of sexual function throughout pregnancy and to try to highlight the predicting factors of sexual dysfunction in pregnant women.

## 2. Materials and Methods

### 2.1. Establishment of the Study Group

The study group consisted of 144 pregnant women, aged 16 to 45. Participants were recruited from the Obstetrics and Gynecology Clinic of SCMUT in Timisoara, Romania, over a period of 12 months. Patients above 16 years of age who agreed to the idea of sexual health evaluation, had a sexual partner at the time of recruitment, had the ability to read and understand Romanian were included in the study. Women with endocrinological, neurological, and psychiatric underlying disorders that could affect sexual functioning, women with severe somatic disorders, women with complications of pregnancy, women known to abuse or depend on psychoactive substances, illiterate women, and those taking medications that could affect sexual functioning were excluded from the study.

In this particular study, the sample selection was conducted using convenience sampling. Due to various practical constraints, such as time limitations and resource availability, we opted for a non-randomized approach. The participants were recruited based on their accessibility and willingness to participate rather than using a systematic randomization process.

Written informed consent was obtained from all participants. The study was conducted according to the guidelines of the Declaration of Helsinki and approved by the Ethics Committee of SCMUT, Timisoara, Romania.

### 2.2. Data Collection and Procedure

Patients were evaluated three times during pregnancy. Their first evaluation was at the time of their first appointment in the Obstetrics and Gynecology Clinic, approximately around weeks 4 to 6 of pregnancy. Gestational age was established from the last menstrual cycle and verified by ultrasound. Patients were asked to fill out a socio-demographic form. The data collected in the socio-demographic form were age, area of residence, education, occupational status, monthly income, marital status, and religion.

A second evaluation was performed at around week 12 of gestation, when patients came in for their routine checkup. The last evaluation was done in the third trimester, around weeks 33–36 of pregnancy.

At the time of each evaluation, patients were asked to fill out the Female Sexual Function Index (FSFI) and the Body Exposure in Sexual Activities Questionnaire (BESAQ). Also, the Body Mass Index (BMI) was assessed at the time of each evaluation. At the time of the last evaluation, we asked patients to also fill in Beck’s Depression Inventory (BDI-II) and a questionnaire regarding psychological aspects of pregnancy and the patients’ relationship with her partner. Based on this questionnaire, we gave a score for the patient’s relationship satisfaction and a score for the patient’s psychological status regarding pregnancy. The score for relationship satisfaction ranges between 4 and 20 points, with higher scores indicating higher relationship satisfaction. The minimum score for the psychological status regarding pregnancy is 0, indicating low levels of adaptation to the physical and psychological changes caused by pregnancy as well as anxiety related to pregnancy and childbirth, while the maximum score is 6, indicating the patient has adjusted well to pregnancy and has not experienced anxious symptoms related to pregnancy and childbirth.

### 2.3. Instruments

#### 2.3.1. FSFI

The FSFI is a self-report questionnaire that has been validated and demonstrated to be a reliable measure of female sexual function [29]. It contains 19 items that examine desire, arousal, lubrication, orgasm, satisfaction, and pain domains. The FSFI was used in many psychometric studies, studies examining female sexual function and dysfunction, epidemiological studies, and clinical studies on treatment interventions for FSD [30]. The full-scale score ranges between 2 and 36 points. A threshold value of ≤26.55 was established for detecting FSD [31].

#### 2.3.2. BESAQ

BESAQ is a validated self-reported 28-item scale that assesses perceptions of body image in sexual contexts and the level of self-consciousness, anxiety, and avoidance to expose aspects of physical appearance during sexual activities [32]. The questionnaire has demonstrated good psychometric properties and has been shown to be a good predictor of sexual functioning [33]. The score ranges between 0 and 4 points [33]. A lower score indicates a higher level of comfort regarding body exposure during sexual activities. Individuals who score lower on this scale may feel less inhibited or more comfortable with body exposure during sexual activities and may have a more satisfying sexual experience. Higher scores show more anxious and avoidant behaviors in relation to sexual activity [32].

#### 2.3.3. BDI-II

The BDI-II is one of the most popular scales for measuring depressive symptoms worldwide [34]. It is a 21-item questionnaire that addresses the most common symptoms of depression: sadness, pessimism, past failure feelings, loss of pleasure, guilty feelings, punishment feelings, self-dislike, self-criticalness, suicidal thoughts, crying, agitation, loss of interest, indecisiveness, worthlessness, loss of energy, changes in sleeping pattern, irritability, changes in appetite, concentration difficulties, and fatigue [35]. Evidence suggests the BDI-II has great psychometric properties in both psychiatric and nonpsychiatric populations [36]. The cut-off scores for the BDI-II were established: <14 for normal; 14–19 for mild depression; 20–28 for moderate depression; and 29–63 for severe depression [36].

### 2.4. Statistical Analysis

Statistical analysis was performed using MedCalc^®^ Statistical Software version 20.106 (MedCalc Software Ltd., Ostend, Belgium) and R programming language version 4.2.2 (31 October 2022 ucrt). Continuous data are described as mean (±standard deviation [SD]) when they are Gaussian distributed, whereas variables without a Gaussian distribution are described as median (interquartile range) values. The normality of the continuous variable distribution was tested using the Shapiro-Wilk test. Nominal data are presented as absolute frequencies and percentages.

The statistical significance of the differences between the groups of pregnant women with and without FSD was assessed using the Student’s *t*-test for populations with a Gaussian distribution, while the Mann–Whitney U test was used for populations without a Gaussian distribution. The statistical significance between percentages was assessed using Pearson’s chi-squared test or Fisher’s exact test. To compare the three evaluations in a repeated-measures design, we conducted the Friedman test.

The impact of one or more factors on the presence of FSD in pregnant women at the third evaluation was assessed with both univariate and multivariate linear and logistic regression models, while the goodness-of-fit was calculated using Tjur’s R2 method, Nagelkerke B, and the Likelihood Ratio Test (LRT). We considered a *p*-value < 0.05 and a confidence level of 0.95 to be significant for estimating intervals.

We conducted a difference of proportion power calculation for the binomial distribution in order to do the sample size calculation for our logistic regression model. We used a significance level of 0.05, a generally used power of 0.8, and, according to Cohen’s guidelines, a large effect size of 0.8 as parameters for our power analysis. The results showed that we need 25 people in each group to achieve the desired power level and that we need to have equally large sample sizes so that the groups are equally representative and the proportions being compared are equally important. In our study, we decided to use sample sizes beyond the minimum required, such as 28 people in the FSD group and 29 people in the control group, increasing the power, enhancing precision, and robustness of our findings.

## 3. Results

The study sample included 144 women aged between 16 and 45 years (mean age of 29.52 ± 5.33 years). More than half of women (62.5%) had a higher education degree and were employed (85.42%). Also, more than half of women (59.7%) had a monthly income below the average monthly salary in Romania. Most of the women in our study group lived in urban areas (63.2%), and almost all of them were married (90.97%). A large percentage of women were Christian Orthodox (81.94%), while 6.25% were Catholic and 6.25% were Protestants. The socio-demographic characteristics of the study group are presented in Table 1.

A large percentage of women had a regular menstrual cycle (77.08%), and almost half of them (48.61%) were using contraceptives. More than half of women did not have previous births (61.11%), while 27.78% of women had previous abortions. Those who had previous births had natural deliveries in 13.19% of cases, caesarean sections in 22.92% of cases, and in 2.78% of cases, they had given birth using both delivery methods. The majority of women obtained their current pregnancy naturally (93.06%), and more than half of them planned their current pregnancy (67.36%). The obstetrical and gynecological characteristics of the study group as well as the characteristics of the current pregnancy are presented in Table 2.

We used several questions to measure the woman’s degree of satisfaction with their current relationship. We observed that more than half of the women included in the study were overall satisfied with their relationship, being pleased with the support received from their partner, the communication between them, and the emotional closeness between them. The percentages obtained after processing the questions are presented in Table 3.

To evaluate the psychological status during pregnancy, several questions were administered during the third evaluation. We observed that most of the women (75.7%) felt that the physical as well as psychological changes associated with the pregnancies allowed them to do what they needed to do in their daily lives. Almost one-third of women (31.3%) were worried about not being able to cope with professional activity and/or household chores during pregnancy. In addition, half of the women were worried about successfully carrying the pregnancy to term. Also, half of the patients were worried about not being able to cope with labor and/or birth.

We observed that women who felt that the physical changes associated with pregnancy did not allow them to do their everyday life activities were equally distributed when considering the presence of FSD, i.e., 12% with FSD vs. 22% without FSD, *p* = 0.26. Also, similar results were observed when considering the psychological changes during pregnancy and the presence of FSD, i.e., 12% with FSD vs. 12% without FSD, *p* = 0.97. At the same time, we did not observe significant differences between the percentages of women worried that they cannot cope with professional activity during pregnancy and the presence of FSD, i.e., 24% with FSD vs. 33% without FSD, *p* = 0.39. When considering the women who were worried about successfully carrying the pregnancy to term, we observed no significant differences between the percentages of women with FSD vs. without FSD, i.e., 44% with FSD vs. 59% without FSD, *p* = 0.17. Similar results were observed when considering the women who were worried that they would not be able to cope with labor and/or birth, namely, no significant percentages between women with FSD and without FSD, i.e., 60% vs. 54%, *p* = 0.57. The administered questions are presented in Table 4.

The BMI increased from the first evaluation to the third one (Table 5), with the median values being statistically significant different at the three evaluations, *p* < 0.001. Median values for BMI-1st evaluation, BMI-2nd evaluation, and BMI-3rd evaluation were 23.71 (20.57–27.37), 24.46 (22.08–28.63) and 28.95 (25.92–32.81), respectively. There were also significant differences between BMI-1st evaluation and BMI-2nd evaluation (*p* < 0.001) and, also, between BMI-1st evaluation and BMI-3rd evaluation (*p* < 0.001), as well as between BMI-2nd evaluation and BMI-3rd evaluation (*p* < 0.001).

When considering the FSFI’s six domains, we observed that lubrication, satisfaction, pain, and the FSFI total score significantly decreased from the 1st to the 3rd evaluation. On the other hand, the score for the orgasm domain increases from the 1st to the 2nd evaluation and decreases from the 2nd to the 3rd evaluation, the differences being statistically significant. We observed no statistically significant difference between evaluations regarding the desire and arousal domains. Similarly, we observed that BESAQ scores showed no statistically significant differences between evaluations. In this analysis, we included only the patients who were sexually active during all three evaluations. The results are shown in Table 5, Figure 1 and Figure 2.

We observed that a percentage of 17.3% of women presented with FSD at the first evaluation, while at the second and third evaluations, the percentage increased to 25.5% and 49.1%, respectively.

We compared the group of women with FSD vs. women without FSD at the third evaluation, considering their socio-demographic characteristics, their obstetrical and gynecological history, body composition, body image, satisfaction with relationships, psychological status regarding pregnancy, and depression level.

We observed no significant differences when considering socio-demographic characteristics and variables related to body composition, body image, and emotional and psychological components. Regarding the obstetrical and gynecological characteristics, we observed significant differences between patients with FSD and patients without FSD in relation to previous abortions (*p* = 0.044). The results are shown in Table 6.

At the third evaluation, we observed that all the underweight women presented FSD (4%), while similar percentages of women with and without FSD presented normal weight. Regarding overweight women, 54% presented FSD, while 38% were without FSD, and regarding obese women, 29% presented FSD, while 52% were without FSD (*p* = 0.28).

We also observed that 88.2% of women did not present depression, while 5.6% presented mild depression and 4.9% presented moderate depression. A percentage of 1.4% of women presented with severe depression. When comparing women with and without FSD, we observed that the percentages of women with depression were similar, except in the case of severe depression, where all the women presented FSD.

We applied univariate linear regression models to assess the influence of measured parameters on the presence of FSD in pregnant women at the third evaluation. The relationship satisfaction score was found to be an independent predictor of FSD during pregnancy (R^2^ = 0.134). The increase in relationship satisfaction score leads to an increase in FSFI total score, the relationship being directly proportional. The increase in relationship satisfaction score by 1 unit results in an increase in the FSFI total score of 1.09. The results are shown in Table 7.

A multivariate logistic regression analysis was performed to assess the factors that influenced the presence of FSD at the third evaluation. The continuous independent variables were found to be linearly related to the logit of the dependent variable.

The resulting model was statistically significant (*p* = 0.007). The Tjur’s R^2^ coefficient was 0.234. The predictor variables explained 23.4% of the variance in the presence of FSD. The results are shown in Table 8.

We observed that the risk of developing FSD increased with a higher BDI score. A 1 unit increase in BDI score increases the risk of developing FSD by 1.15 times. The presence of abortion in the patient’s history leads to an increase in the risk of developing FSD by 5.96 times. Higher BMIs were found to be a protective factor against FSD. A 1 unit increase in BMI (kg/m^2^) decreases the odds of developing FSD by 0.79 times. Being unmarried was also found to be a protective factor against FSD, decreasing the odds of developing FSD by 0.09.

## 4. Discussion

Sexual function is an important aspect of overall physical and emotional well-being, and changes in sexual function during pregnancy can have a significant impact on a woman’s quality of life. It is also an important aspect of the relationship between partners. Changes in sexual function can affect communication, intimacy, and overall relationship satisfaction. Therefore, addressing sexual dysfunction during pregnancy is important for maintaining healthy relationships [18].

The prevalence of sexual dysfunction during pregnancy varies depending on the specific type of dysfunction and the population being studied. However, research suggests that sexual dysfunction is common during pregnancy, with some studies reporting rates of up to 80% for sexual problems such as decreased libido, difficulty becoming aroused, and pain during intercourse [37]. Factors that can contribute to sexual dysfunction during pregnancy include physical changes, hormonal changes, psychological factors, and relationship issues [38]. Our results are comparable with previous studies in the field, showing that a percentage of 49.1% of women included in the study presented with FSD at some point during pregnancy.

Our results provide valuable insights into the changes observed in various parameters and scores related to sexual function and body exposure during pregnancy. These results shed light on the potential impact of pregnancy on women’s physical and sexual well-being. We observed that certain domains of the FSFI exhibited statistically significant differences across the three evaluations. Lubrication, orgasm, satisfaction, pain, and the FSFI total score demonstrated changes over time. These findings indicate that there may be variations in sexual experiences and functioning during pregnancy. It’s important for healthcare professionals to be aware of these potential changes and address any concerns or challenges faced by pregnant women in their sexual lives [18].

Interestingly, the desire and arousal domains of the FSFI scores did not show statistically significant differences across the evaluations. This suggests that while other aspects of sexual function may fluctuate, women’s desire for and arousal during sexual activity may remain relatively stable during pregnancy. However, it is important to note that individual experiences and preferences may vary, and open dialogue between partners can help address any changes or concerns in these areas [37].

Contrary to expectations, our study found a decrease in the lubrication domain of sexual function from the first to the third evaluation during pregnancy, indicating challenges for pregnant women in sexual activity as pregnancy progresses [15]. This suggests that while hormonal factors are important, subjective factors may have a greater influence on sexual function. It is possible that the cognitive component of arousal takes precedence over the reflex hormone-dependent response, contributing to these findings. These results align with previous studies highlighting the multifaceted nature of pregnancy and postpartum sexuality, where bio-psycho-social factors often have a greater impact than physical factors like vaginal trauma or breastfeeding [39].

Our study found variations in sexual experiences related to orgasm throughout pregnancy. Women reported slightly higher FSFI scores for the orgasm domain at the second evaluation compared to the first, indicating a potential improvement or maintenance of orgasmic function during the first pregnancy period. However, at the third evaluation, the FSFI scores decreased, indicating a potential decline in orgasmic function during the late stages of pregnancy. Hormonally, the elevation in estrogen levels has been linked to heightened sexual sensitivity and improved genital response, potentially intensifying orgasms. Conversely, the rise in progesterone production during pregnancy could impact sexual desire and orgasmic experiences, implying that hormonal changes alone may not fully account for the enhancement of orgasmic function in pregnant women [40]. Emotional well-being and the positive emotions that women feel at the beginning of pregnancy can positively affect sexual function and relationship dynamics. Pregnancy deepens the emotional connection, bonding, and intimacy between partners, creating a supportive and nurturing environment that promotes sexual satisfaction and orgasmic function.

The decrease in the orgasm domain scores at the end of the pregnancy might as well be attributed to psychological factors, including body image concerns, stress, anxiety, and mood changes. Throughout pregnancy, women may experience fluctuations in body image due to weight gain and bodily changes, which can impact their self-esteem and sexual confidence. Stress, anxiety, and physical discomfort during pregnancy can impact sexual function. Psychological factors like stress and anxiety may contribute to variations in orgasm domain scores throughout pregnancy. Additionally, the physical discomfort caused by the growing fetus, weight gain, and changes in body shape can affect sexual experiences, including orgasm. Discomfort and concerns about physical changes may further lead to decreased sexual satisfaction and orgasmic function as pregnancy progresses [11,16,17].

Sexual satisfaction was also significantly affected during pregnancy. Our findings align with previous studies that demonstrated that sexual satisfaction significantly declined during pregnancy [4].

Regarding the pain domain, we observed that it was significantly affected during pregnancy, suggesting that women may experience an increase in sexual pain or discomfort, which can be attributed to physiological changes, heightened sensitivity, or pelvic pressure, as well as to the decrease in lubrication [41].

The BESAQ score, which measures body exposure in sexual activities, did not exhibit statistically significant differences across the evaluations. This suggests that the comfort level and body exposure to sexual activities may remain relatively consistent during pregnancy. However, it is crucial to recognize that body image concerns and self-esteem may still play a role in sexual well-being during this period. Supportive discussions and interventions targeting body image issues can help women maintain a positive self-perception and enhance their sexual experiences [32,33].

We performed a more thorough assessment of sexual function in the third trimester of pregnancy because sexual dysfunction during the third trimester has been associated with a negative impact on postpartum sexual function and recovery [42]. As expected, our results showed that low satisfaction with the relationship is an independent predictor of developing FSD during pregnancy. Previous studies also highlighted the importance of a qualitative couple’s relationship in relation to sexual functioning and well-being [43]. The presence of clinical symptoms of depression and a history of abortion were found to be independent risk factors, increasing the odds of developing FSD during the third trimester of pregnancy.

The link between depression and sexual dysfunction has been well studied. Depression can cause a decrease in sexual desire, reducing interest in sexual activity. Pregnancy often brings physical changes and increased demands on the body, leading to fatigue and exhaustion. Depression can amplify feelings of fatigue, making it challenging to engage in sexual activity and contributing to sexual dysfunction. Depression is often accompanied by anxiety, and during pregnancy, this anxiety may revolve around concerns for the health of the baby, childbirth, or parenting [44]. These worries can distract from sexual intimacy and contribute to sexual dysfunction. Moreover, depression can also strain relationships and lead to emotional distance or communication difficulties between partners. This strain can extend to the sexual aspect of the relationship, creating barriers to intimacy and resulting in sexual dysfunction during pregnancy [45].

The literature on the relationship between a history of abortion and sexual dysfunction is limited. However, several potential explanations exist for the findings. Emotional and psychological effects, such as guilt, shame, or anxiety, may arise from a past abortion and impact a woman’s sexual well-being during subsequent pregnancies. Additionally, a history of abortion can affect relationship dynamics, leading to communication difficulties, trust issues, or emotional distance, which can contribute to sexual problems. Physical trauma or complications from previous abortions may also cause discomfort, pain, or reduced sexual functioning in subsequent pregnancies. Cultural, societal, and religious factors can influence attitudes towards abortion and, consequently, impact sexual dynamics and contribute to sexual dysfunction during pregnancy [43,46].

As previous studies suggested, risk factors that influence the development of FSD during pregnancy will continue to do so in the postpartum period [42]. Addressing sexual dysfunction during pregnancy can help prevent postpartum sexual dysfunction and promote recovery after childbirth.

The surprising finding in our study is that a higher BMI was found to be a protective factor against the development of sexual dysfunction in the third trimester of pregnancy. Contrary to expectations based on the general relationship between BMI and sexual function, our results show that higher BMIs significantly decrease the likelihood of experiencing sexual dysfunction during this stage of pregnancy. Previous literature has consistently shown that increasing BMI is associated with decreased sexual desire, arousal, body image concerns, and physical discomfort or limitations, which can impact sexual activity and function. It is intriguing that this relationship between BMI and sexual function appears only at the end of pregnancy, while throughout the course of pregnancy, an increase in BMI is associated with sexual dysfunction. A potential explanation for this discrepancy could be the influence of the emotional aspect of sexuality among overweight or obese women. Interestingly, a study examining women’s choices of sexual response models found that overweight or obese women were more inclined to choose the Basson model, which emphasizes emotional connection and intimacy in sexual experiences [47]. This suggests that the emotional component of sexuality might play a significant role in the sexual well-being of overweight or obese pregnant women, potentially contributing to the observed protective effect of a higher BMI against sexual dysfunction. This finding further highlights the paradoxical nature of sexual dysfunction observed during pregnancy. However, it is important to acknowledge the limitations of our convenience sample and the need for further research with larger and more diverse populations to fully understand the complexities of this relationship and the potential moderating effects of sample characteristics [48,49].

One of the major strengths of our study is its prospective design, which allowed us to follow women longitudinally throughout their pregnancies. However, it is important to acknowledge some limitations. Firstly, our sample was recruited from a single clinic, which may limit the generalizability of our findings. Additionally, the sample size was relatively small, which could impact the statistical power of our results. The participants in our study were recruited through convenience sampling, which introduces the possibility of selection bias. As a result, caution should be exercised when extrapolating our results to a broader population. Future research utilizing random sampling techniques would be valuable in providing a more representative sample and enhancing the external validity of the findings.

## 5. Conclusions

Our study revealed a decline in female sexual function over the course of pregnancy, specifically in the domains of lubrication, satisfaction, and pain. Regarding the predicting factors of FSD and risk factors, we identified low satisfaction with the relationship as an independent predictor of developing FSD during pregnancy. This highlights the critical role of relationship quality in sexual functioning and overall well-being. Additionally, the presence of clinical symptoms of depression emerged as an independent risk factor, increasing the likelihood of developing FSD in the third trimester. The history of abortion is associated with an increased risk of FSD during pregnancy. Psychological effects, relationship dynamics, physical trauma, cultural or religious beliefs, fear, and anxiety surrounding previous abortion experiences can impact sexual well-being during subsequent pregnancies. Interestingly, a higher BMI unexpectedly acted as a protective factor against sexual dysfunction in the third trimester of pregnancy. This finding contradicts the general relationship between BMI and sexual function and warrants further investigation to better understand this paradoxical result.

In our title, we highlight a paradox concerning the contrasting influences on sexual function during pregnancy. Specifically, there is a divergence between the physical and hormonal aspects, which we would expect to enhance sexual function, and the subjective factors. These subjective factors, encompassing relationship satisfaction, psychological well-being, and emotional aspects of sexuality, can actually exert a more significant impact on sexual function during pregnancy, thereby suggesting their heightened influence.

Overall, the findings suggest that sexual function during pregnancy is multifaceted and influenced by various physical, psychological, and relational factors. This study emphasizes the importance of addressing factors such as relationship satisfaction, depression, history of abortion, and body composition in understanding and managing sexual dysfunction during pregnancy. By addressing these factors, healthcare professionals can potentially prevent postpartum sexual dysfunction and promote recovery after childbirth. Further research is required to explore the complexities of these relationships and provide a deeper understanding of sexual function during pregnancy.

## Figures and Tables

**Figure 1 healthcare-11-01914-f001:**
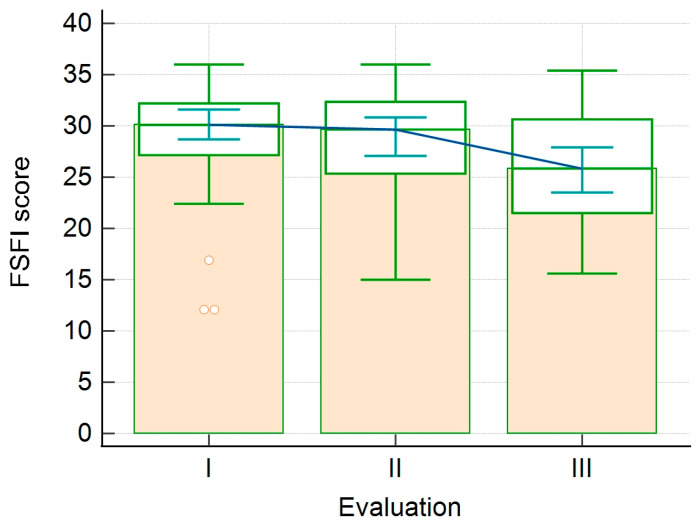
Evolution of the FSFI score between evaluations (I—1st evaluation; II—2nd evaluation; III—3rd evaluation).

**Figure 2 healthcare-11-01914-f002:**
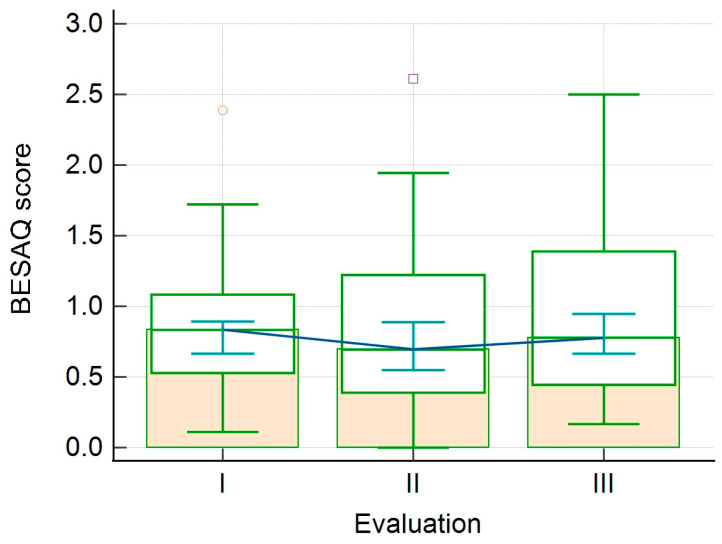
Evolution of the BESAQ score between evaluations (I—1st evaluation; II—2nd evaluation; III—3rd evaluation).

**Table 1 healthcare-11-01914-t001:** Socio-demographic characteristics of the study group.

Socio-Demographic Data	*n* = 144
Age [Years]	29.52 (±5.33)
Education	
Elementary education	6 (4.17%)
Secondary education	48 (33.33%)
Higher education	90 (62.5%)
Professional status	
Unemployed	21 (14.58%)
Employed	123 (85.42%)
Income	
Under the average monthly salary	86 (59.72%)
Average monthly salary or above	58 (40.28%)
Living area	
Urban	91 (63.2%)
Rural	53 (36.8%)
Civil status	
Married	131 (90.97%)
Unmarried	13 (9.03%)
Religion	
Orthodox	118 (81.94%)
Catholic	9 (6.25%)
Protestant	9 (6.25%)
Others	8 (5.55%)

Notes: Continuous variables (with Gaussian distribution) are indicated as mean (standard deviation), while continuous variables (with non-Gaussian distribution) are indicated as median (interquartile range), and categorical variables are indicated as percentage (absolute frequency) in the sample.

**Table 2 healthcare-11-01914-t002:** Obstetrical and gynecological characteristics of the study group.

Obstetrical-Gynecological Data	*n* = 144
Menarche [Years]	13.00 (12.00–14.00)
Menstrual cycle [Regular]	111 (77.08%)
Previous births [Yes]	56 (38.89%)
Previous birth method	
Natural	19 (13.19%)
Caesarean section	33 (22.92%)
Both methods	4 (2.78%)
No previous births	88 (61.11%)
Previous abortion [Yes]	40 (27.78%)
Contraceptive use	
>5 years	14 (9.72%)
1–5 years	22 (15.28%)
<1 year	34 (23.61%)
No	74 (51.39%)
Planned pregnancy [Yes]	97 (67.36%)
Period trying conception	
Unplanned	47 (32.63%)
<6 months	55 (38.18%)
6 months–1 year	16 (11.1%)
>1 year	26 (18.1%)
Method obtaining actual pregnancy	
Natural	134 (93.06%)
Assisted human reproduction techniques	10 (6.94%)

Notes: Continuous variables (with Gaussian distribution) are indicated as mean (standard deviation), while continuous variables (with non-Gaussian distribution) are indicated as median (interquartile range), and categorical variables are indicated as percentage (absolute frequency) in the sample.

**Table 3 healthcare-11-01914-t003:** Satisfaction with the current relationship.

Question	1	2	3	4	5
How satisfied are you overall with your current relationship?	4 (2.8%)	1 (0.7%)	4 (2.8%)	25 (17.4%)	110 (76.4%)
How satisfied are you with the support you receive from your partner?	3 (2.1%)	1 (0.7%)	3 (2.1%)	31 (21.5%)	106 (73.6%)
How satisfied are you with the communication you have with your partner?	4 (2.8%)	0 (0%)	5 (3.5%)	46 (31.9%)	89 (61.8%)
How satisfied are you with the degree of emotional closeness between you and your partner?	3 (2.1%)	3 (2.1%)	4 (2.8%)	42 (29.2%)	92 (63.9%)

Notes: 1 = very unpleased; 2 = unpleased; 3 = neutral; 4 = pleased; 5 = extremely pleased.

**Table 4 healthcare-11-01914-t004:** Psychological status during pregnancy.

Question	Answer [Yes]
Were/are you satisfied with how you were able to adapt to this pregnancy overall?	133 (92.4%)
Do you feel that the physical changes associated with this pregnancy have allowed you to do what you needed to in your daily life?	115 (79.9%)
Do you feel that the psychological changes associated with this pregnancy allowed you to do what you needed to in your daily life?	127 (88.2%)
Have you been/are you worried that you cannot or will not be able to cope with professional activity and/or household chores during pregnancy?	45 (31.3%)
Have you been/are you worried about successfully carrying your pregnancy to term?	81 (56.3%)
Were/are you worried that you will not be able to cope with labor and/or birth?	79 (54.9%)

Notes: For each question we indicated the number of Yes answers as well as the corresponding percentage.

**Table 5 healthcare-11-01914-t005:** Comparison results at the three evaluations during pregnancy.

Parameters	Evaluation I (N = 45)	Evaluation II (N = 45)	Evaluation III (N = 45)	*p*-Value
BMI	23.71 (20.57–27.37)	24.46 (22.08–28.63)	28.95 (25.92–32.81)	<0.001 *
FSFI score				
Desire	4.20 (3.60–4.80)	4.20 (3.60–5.40)	3.60 (2.40–4.80)	0.16
Arousal	5.10 (4.43–5.40)	5.40 (4.20–5.70)	4.50 (3.60–5.70)	0.06
Lubrication	5.10 (4.80–6.00)	5.10 (4.50–6.00)	4.80 (3.60–5.40)	0.02 *
Orgasm	4.80 (4.40–5.60)	5.20 (4.40–6.00)	4.40 (3.60–5.20)	0.04 *
Satisfaction	6.00 (5.20–6.00)	6.00 (4.80–6.00)	4.80 (4.40–6.00)	0.002 *
Pain	5.60 (4.40–6.00)	5.60 (4.40–6.00)	4.40 (3.60–6.00)	<0.001 *
Total	30.60 (27.68–32.70)	30.50 (26.48–33.45)	27.70 (22.10–30.30)	0.01 *
BESAQ score	0.72 (0.38–1.11)	0.56 (0.27–0.88)	0.78 (0.38–1.33)	0.28

Abbreviations: BMI—Body Mass Index; FSFI—Female Sexual Function Index; BESAQ—Body Exposure in Sexual Activities Questionnaire. Notes: * Difference between evaluations is significant (*p* < 0.05). The data were presented as the median (interquartile range).

**Table 6 healthcare-11-01914-t006:** Comparative description of women with FSD vs. women without FSD at the third evaluation.

Parameters ^a^	With FSD (*n* = 28)	Without FSD (*n* = 29)	*p*-Value ^b^
**Socio-demographic data**			
Age	28.29 (±5.74)	27.17 (±4.09)	0.428
Education [Higher education]	17 (60.71%)	19 (65.52%)	0.929
Civil status [Married]	26 (92.85%)	25 (86.20%)	0.413
Professional status [Employed]	23 (82.14%)	23 (79.31%)	0.786
Income [Average monthly salary or above]	12 (42.86%)	12 (41.38%)	0.910
Living area [Urban]	18 (64.29%)	19 (65.52%)	0.922
Religion [Christian orthodox]	22 (78.57%)	22 (75.86%)	0.723
**Obstetrical and gynecological characteristics**			
Menarche [Years]	13.24 (±1.85)	13.24 (±1.55)	0.221
Menstrual cycle [Regular]	23 (82.14%)	22 (75.86%)	0.561
Previous births [Yes]	12 (42.86%)	8 (27.59%)	0.227
Previous abortion [Yes]	9 (32.14%)	3 (10.34%)	0.044 *
Planed pregnancy [Yes]	15 (53.57%)	20 (68.96%)	0.233
**Body image, psychological and emotional components**			
BMI [kg/m^2^]	28.21 (±4.20)	30.19 (±4.73)	0.160
BESAQ score	0.86 (0.44–1.39)	0.72 (0.39–0.94)	0.400
BDI score	4.50 (2.75–9.75)	3.00 (2.00–7.00)	0.248
Satisfaction with relationship score	20.00 (17.00–20.00)	20.00 (19.00–20.00)	0.090
Psychological status regarding pregnancy score	5.00 (3.00–6.00)	4.00 (4.00–5.00)	0.329

Abbreviations: FSD—Female Sexual Disfunction; BMI—Body Mass Index; BESAQ—Body Exposure in Sexual Activities Questionnaire; BDI—Beck’s Depression Inventory. Notes: * The difference between variables is significant (*p* < 0.05). ^a^ Continuous variables (with Gaussian distribution) are indicated as mean (±std.dev.), while continuous variables (with non-Gaussian distribution) are indicated as median (interquartile range), and categorical variables as percentage (absolute frequency) in the sample. ^b^ *p*-value was computed by the Mann–Whitney *U* test for continuous variables (with a non-Gaussian distribution) and Pearson’s chi-squared (or Fisher’s exact) test for nominal variables.

**Table 7 healthcare-11-01914-t007:** Independent predictors for FSD in pregnant women at the third evaluation (univariate linear regression model; R^2^ = 0.134).

Predictor Variable	Estimates	95% CI		*p*-Value
		Lower	Upper	
Relationship satisfaction score	1.09	0.34	1.83	0.005 *

Notes: * The predictor variable is significant.

**Table 8 healthcare-11-01914-t008:** Independent risk and protective factors for FSD in pregnant women at the third evaluation (multivariate logistic regression model; Tjur’s R^2^ = 0.234; Nagelkerke = 0.328; LRT, *p*-value = 0.004).

Predictor Variable	OR	95% CI		*p*-Value
		Lower	Upper	
Civil status [Unmarried]	0.09	0.01	0.83	0.045 *
BMI [kg/m^2^]	0.79	0.64	0.93	0.012 *
BDI score	1.15	1.03	1.35	0.037 *
Previous abortion [Yes]	5.96	1.34	34.79	0.028 *

Notes: * The predictor variable is significant both independently and as a co-factor. Abbreviations: BMI, body mass index; BDI, Beck’s Depression Inventory; LRT, Likelihood Ratio Test.

## Data Availability

The data presented in this study are available upon request from the corresponding authors. The data are not publicly available due to reasons concerning the privacy of the subjects.

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
