# Peer review of "The Paradox of Sexual Dysfunction Observed during Pregnancy"

_healthcare, 2023, doi:10.3390/healthcare11131914_

Round 1

Reviewer 1 Report (Previous Reviewer 1)

Dear authors thank you for addressing the issues that have been brought to your attention and for your hard effort revising your manuscript. I wish you all the best to your upcoming projects.

Author Response

Reviewer 2 Report (New Reviewer)

This paper is dealing with interesting topic, paper is nicely written, and it implies that topic of womens sexuality during pregnancy and postpartum is complex problem with many sides to look and research. 

I reccoment detail check of the English language. 

Author Response

Reviewer 3 Report (New Reviewer)

Thank you for your work, next I will proceed to indicate some important corrections about your work.

ABSTRACT

1. Do not use abbreviations without first describing what they mean.

2. The abstract is longer than allowed. Check the word count and summarize it.

METHODOLOGY

3. How was the study sample selected? Was any type of system followed to randomize the sample? Is it consecutive and convenience sampling? Detail in the manuscript how the sample selection was done.

4. Have you done a sample calculation? Detail it.

RESULTS

5. There is no need to add methodological aspects previously explained in the results. Remove sentences 428-431, 526-529 and 1087-1093.

6. The results reflected in lines 577-590 are difficult to interpret. Add a table reflecting the p value. Also, be careful when writing the results when these have a p value greater than 0.05.

7. Do not make value judgments when exposing the results. Review line 711.

8. Introduce the abbreviations of the tables within them and not in the manuscript.

9. Include the logistic regression table in the manuscript, indicating the adjusted R2 and B, as well as the degree of significance.

DISCUSSION

10. There are paragraphs without citing. Review the discussion and add the relevant references.

11. Rewrite the discussion, there are excessively long paragraphs that make reading difficult (for example from 1454-1484).

12. Rewrite the discussion, in it the results are presented again but there is hardly any discussion with the existing bibliography.

13. Do not repeat the same reference 41 twice in a row.

14. Rewrite the limitations of the study. They are not clear.

CONCLUSIONS

15. Rewrite the conclusions. At the beginning of the conclusions, the objective of the research should be answered.

Round 2

Reviewer 3 Report (New Reviewer)

Dear authors, I am providing you with further revisions that were not clear in the first review. In the right margin of the manuscript, you can see the lines I am referring to from the previous review. They do not correspond to the usual text lines. Please download the PDF version of the manuscript to follow the instructions. It is urgent that you adhere to the publication guidelines regarding word limits for sections, font sizes, line spacing, and the formatting of tables and figures. Otherwise, I will be forced to reject the article.

ABSTRACT

1. The maximum word count exceeds the allowed limit. Additionally, the publication guidelines state that the abstract should be structured. Please adhere to the publication guidelines.

2. The statement found between lines 32 and 35 of the manuscript, "These findings support previous studies indicating that pregnancy and postpartum sexuality are multifaceted phenomena, and that bio-psycho-social factors often have a greater impact on sexuality than the commonly studied physical factors," yields ambiguous conclusions. Do they sometimes have a greater impact? In what instances? In what instances do they not? How much greater?

METHODOLOGY

3. Add to the manuscript the type of sampling employed, indicating that it was not randomized sampling but convenience sampling.

4. Conduct a sample size calculation and include it in the manuscript. When using logistic regressions and predicting a model, we need to justify that our sample is representative of the model we want to predict.

RESULTS

5. Regarding the tables, if you do not wish to include the notes within them (which is recommended), at least make them smaller than the manuscript text size to avoid confusion with the manuscript text and potential errors for readers. Also, do not increase the line spacing and adhere to the publication guidelines; these errors are highly visible, as in Table 4.

6. Do not include methodological and statistical test aspects in the results section that have already been explained in the methodology section. Remove them.

7. The results reflected in lines 239-252 are difficult to interpret. Add a table reflecting the p-value. Also, be careful when writing the results when the p-value is greater than 0.05.

8. Add a goodness-of-fit test (Hosmer and Lemeshow test) to the logistic regression table.

DISCUSSION

9. Why do you think this? "The surprising finding in our study is that a higher BMI was found to be a protective factor against the development of sexual dysfunction in the third trimester of pregnancy." What makes you think this is a novel finding in the literature? Does it relate to the sample selection?

10. Within the study's limitations, you did not mention the non-randomization of the sample or the limitations in data collection for various variables such as sexual relationship. Please complete the limitations section and provide future lines of research.

CONCLUSIONS

Please rewrite the conclusions of your study.

11. The first sentence of the conclusions should be removed as it is not accurate. In the discussion section, highlight new findings such as the one related to BMI and female sexual dysfunction.

12. You cannot start a conclusion like this. Conclusions should begin by addressing the research objective, which is different from what you have done by restating the research objective

Author Response

This manuscript is a resubmission of an earlier submission. The following is a list of the peer review reports and author responses from that submission.

Round 1

Reviewer 1 Report

Dear authors congratulations for your hard work. It is true that female sexual dysfunction is a neglected subject in comparison with erectile dysfunction or ejaculation disorders, especially when we are dealing with pregnancy. Many can easily blame the hormons but this isn't the case. Besides the psychological component, anatomical and physical issues can lead to FSD during pregnancy. 

Your study pretty much verifies what we know or we suspect about FSD in pregnancy. If I may I would like to point out some things regarding your study:

1.Would't it be better if you had evaluated the participants at smaller intervals, for instance if you had it at about 20-24 weeks time of gestation?

2. Wouldn't it be better if you had use a validated questionnaire to value the relationship satisfaction?

3. Have it crossed your mind to use a visual scale of evaluating pain/dyspareunia during intercourse?

4. It was very interesting that you haven't noticed any association between  depression development of FSD.

5. It is strange that you have found a continues decline at FSFI scores since most of the studies describe an increase in libido, improvement of vaginal lubrication and a better adaptation of women to their status at second semester.  

6. You haven't provided any information about any other childbirths, if the women participating had other children to take care of, if they had help at home, or perhaps the impact of nausea and urgency to sexuality.

7. At the end of the abstract you comment and I quote "These findings support previous studies that indicate that pregnancy and postpartum sexuality is a multifaceted phenomena and bio-psycho-social factors sometimes impact sexuality  more that the most frequent studied physical factors". How does your study supports findings about postpartum sexuality, since you haven't evaluated women beyond pregnancy?

8. My last remark has to do with your title: where is the paradox? You haven't addressed that issue!

Reviewer 2 Report

I have read the manuscript. It is an interesting work, but there are little issues that need to be clarified. Some comments:

·         The format of the paper should be revised. For example: (1) units should be always inserted, (2) title of paragraph “2.2 Method” is too generic, (3) Acronyms should be defined before their first use, (4) tables should be better formatted.

·         In figures, the classes are not defined. It is difficult to interpret the presented classes.

·         Are there any correlations between the anamnestic data and the obtained results? I suggest performing a correlation analysis.

Reviewer 3 Report

The objectives of the manuscript are genuine. Little is known regarding the human sexuality during pregnancy. The findings are interesting and would be a good addition to the literature.

The only concern is that the manuscript is too long with repetitions at different sections. Most of the repetitions appeared in the discussion and conclusion (which is also too long).  

In my opinion, making the text shorter and omitting repeated paragraphs along with English language edition would make a good article.